

Hygroscopic growth effect on aerosol light scattering in the urban area of
Beijing: a long-term measurement by a wide-range and high-resolution
humidified nephelometer system
**Pusheng Zhao[1*], Jing Ding[2, 1], Xiang Du[2, 1], and Jie Su[1]**
[1] Institute of Urban Meteorology, China Meteorological Administration, Beijing 100089, China
[2] State Environmental Protection Key Laboratory of Urban Ambient Air Particulate Matter Pollution
Prevention and Control, College of Environmental Science and Engineering, Nankai University,
Tianjin 300071, China
* *Correspondence to:* P. S. Zhao (pszhao@ium.cn)
**Abstract:**
Hygroscopicity is an important feature of ambient aerosols, which is very crucial to the
study of light extinction, radiation force, and formation mechanism. The light scattering
hygroscopic growth factor ($f$(RH)) is an important parameter which is usually measured
by the humidified nephelometer system and could better describe the aerosol
hygroscopicity under wide particle size range and continuous relative humidity (RH).
The $f$(RH) can be applied to the establishment of a parameterization scheme for light
extinction, the calculation of hygroscopicity parameter ($\kappa$), and also the estimation of
aerosol liquid water content (ALWC). However, the humidified nephelometer system
in the previous studies could only observe the $f$(RH) below 90% due to the larger error
of the sensor under high RH (>90%). Furthermore, the $f$(RH) observations in North
China Plain needs to be greatly strengthened both in the temporal resolution and the
observation duration. In view of this, an improved high-resolution humidified
nephelometer system was established to observe the $f$(RH) of $PM_{2.5}$ for a wide RH range
between 30%-96% in the urban area of Beijing over three seasons (winter, summer, and
autumn) in 2017. It was found that the $f$(80%) at 525nm of $PM_{2.5}$ was evidently higher
under the polluted conditions and highly correlated with the fractions of all the water-
soluble ions. A two-parameter fit equation was selected to fit the observed $f$(RH) data.
For each season, the fitting curve under the very clean condition was lower than that of
other conditions. And the $f$(RH) points of polluted conditions were more concentrated
with higher fitting $R^2$ for summer and autumn data. The hygroscopicity of aerosol under
higher RH was probably enhanced when compared with the data in the previous study
conducted in NCP. In summer, the fitting $f$(RH) showed a significant dependence on
wavelength for each pollution condition. However, there was an opposite performance
in the $f$(RH) curves of different wavelengths for the very clean condition in winter. It
was shown that The simulation showed that the maximum uncertainty of $f$(RH) was
less than 10%.



## 1. Introduction

Atmospheric aerosols influence the atmospheric visibility and earth-atmosphere radiation budget directly by scattering and absorbing solar radiation (Charlson et al., 1992; Schwartz et al., 1995; DeBell et al., 2006), and indirectly by modifying microphysical properties and lifetime of clouds (Twomey, 1974; Albrecht, 1989; Rosenfeld, 2000). The particle size, single scattering albedo (SSA), asymmetry parameter (AP), and refractive index are the key parameters in estimating the aerosol radiative forcing; and these parameters are strongly dependent on the relative humidity (RH). Water uptake can result in enlarged particle sizes, SSA, and AP, change the scattering phase functions, and decreased the refractive index (Cheng et al., 2008). Moreover, numerous studies demonstrate that reaction in aerosol liquid water is an important pathway of secondary aerosol formation, and thus plays a significant role in the overall aerosol chemical composition (McMurry and Wilson, 1983; Ravishankara 1997; Kolb et al., 2010; Jang et al., 2002; Liu et al., 2012; Cheng et al., 2016). Furthermore, water uptake is also important in the remote-sensing measurements or satellite retrievals of aerosol optical properties (Wang and Martin, 2007; Zieger et al., 2012).

Particle size and chemical composition are the main factors affecting the moisture absorption ability of aerosol particles (Köhler, 1936; Hinds, 2011; Petters and Kreidenweis, 2007; Seinfeld and Pandis, 2016). Particle size mainly affects the extent of the Kelvin effect, while the chemical composition is the most crucial factor. Some simulations demonstrated that the changes of aerosol size distribution could only lead to slight variations in hygroscopic growth factor if the chemical compositions were fixed (Fierz-Schmidhauser et al., 2010; Liu 2015; Kuang et al., 2017).

Generally, aerosol hygroscopicity could be described by the diameter growth factor ($g$(RH)), light scattering hygroscopic growth factor ($f$(RH)), and hygroscopicity parameter ($\kappa$). Size-resolved $g$(RH) could be directly measured by the Humidified Tandem Differential Mobility Analyzer (HTDMA) (Liu et al. 1978; Swietlicki et al., 2008). Due to the limitation of the HTDMA itself, only the particles with the dry diameter under 350nm are usually observed. The $f$(RH) is calculated as the ratio of the scattering coefficient at a certain RH to the corresponding dry scattering coefficient (reference RH<40%), which can be measured by a dual nephelometer system (Covert et al., 1972; Rood et al., 1985). In recent years, the $\kappa$-Köhler theory has been widely used to describe the hygroscopic properties of aerosols (Petters and Kreidenweis, 2007), in which all of the chemical composition dependent variables were merged into a hygroscopicity parameter ($\kappa$). It facilitates the intercomparison of particle hygroscopicity obtained by different equipment or different relative humidity. It is applicable to both the single-component and the multicomponent aerosol particles. Knowing the $\kappa$, dry diameter, and the relative humidity (RH), the $g$(RH) and liquid water content (LWC) of aerosol particles could be calculated by using the $\kappa$-Köhler equation.

The $f$(RH) synthetically reflects the impact of water uptake on aerosol particle size, morphology, refractive index, etc., and consequently, on the scattering coefficient. Firstly, the $f$(RH) results can be directly applied to the establishment of a



parameterization scheme for light extinction. The parameterization scheme could be built based on chemical compositions (DeBell et al., 2006; Pitchord et al., 2007), mass concentrations (Chen et al., 2015), or volume concentrations (Chen et al., 2012, 2014). No matter which scheme is used, the effect of relative humidity on extinction or scattering coefficient must be taken into account. Secondly, the $f$(RH) could be used to calculate the hygroscopicity parameter ($\kappa$) combining with the measurement of particle number concentration size distributions (PNSD) (Chen et al., 2014; Kuang et al., 2017). This $\kappa$ does not target specific particle size but represents the overall hygroscopicity of the observed aerosol. Thirdly, the $f$(RH) results derived from a three-wavelength nephelometer system have also been used for the calculation of aerosol liquid water content (ALWC) (Kuang et al., 2018) and number concentrations of cloud condensation nuclei (Tao et al., 2018).

The humidified nephelometer system was firstly used by Pilat and Charlson (1966) to measure the effect of humidity on the light scattering and the size of NaCl particles. Covert et al. (1972) then used a similar system to study the hygroscopic and/or deliquescence effects which were dependent upon relative humidity for pure particles. Rood et al. (1985) improved the humidified nephelometers, which could control the RH more precisely. In the past two decades, Carrico et al. (2000), Day et al. (2000), Koloutsou-Vakakis et al. (2001), Fierz-Schmidhauser et al. (2010), Liu et al. (2016), etc. further improved the humidified nephelometers to accurately measure the $f$(RH) and study the light scattering enhancement characteristics of aerosol. While all the humidified nephelometer system in the previous studies could only observe the $f$(RH) below 90% RH because there would be a ±5% error in the RH measurement in the optical chamber of nephelometer when the RH was humidified over 90%. However, for a hygroscopic particle, its particle size or scattering cross section will increase sharply when the relative humidity exceeds 90%. Accordingly, the results above 90% RH would be quite important for $f$(RH) curve fitting and light scattering calculating.

As we all know, the North China Plain (NCP) is the most severe area of air pollution in China. The aerosol hygroscopicity is an important basis for the studies of atmospheric visibility, radiative forcing, and aerosol secondary formation in this area. The HTDMA had been used in some campaigns in NCP (Massling et al., 2009; Meier et al., 2009; Liu et al., 2011; Wang et al., 2018). As mentioned above, the $g$(RH) observations using HTDMA had been basically limited to 350 nm, and could not be set at multiple RH points. In comparison, the overall hygroscopicity of ambient aerosol at several continuous RH points could be obtained by the $f$(RH) observation, which would also be used to further calculate the hygroscopicity parameter and ALWC. In recent years, the $f$(RH) observation has been carried out in only a few studies in NCP (Yan et al., 2009; Pan et al., 2009; Chen et al., 2014; Kuang et al., 2017). In these studies, either the temporal resolution of $f$(RH) was low, or the observation period was short. Moreover, the $f$(RH) was limited to 90% RH in all these studies.

At present, some numerical pollution prediction systems have been established in the region of NCP. However, the prediction of visibility is short of good means and methods. Thus, an accurate, easy to use, and seasonal representative visibility parameterization scheme is very critical and urgent. Furthermore, owing to the significant changes in the





pollution sources, the physical and chemical characteristics of aerosols in NCP have
also changed evidently in recent years. Above all, the research on aerosol
hygroscopicity is still not enough for NCP. Further study is required, especially in the
area of $f$(RH). In this study, an improved high-resolution humidified nephelometer
system was used to observe the $f$(RH) of $PM_{2.5}$ for a wide RH range between 30%-96%
in the urban area of Beijing over three seasons (winter, summer, and autumn). The main
objectives of this article are to characterize the variations of $f$(RH) and other optical
parameters under different seasons and different pollution levels, and set up the optimal
expressions of $f$(RH).

**2. Instruments and methods**

**2.1 Observation site**

The measurement campaign was performed at the Institute of Urban Meteorological
in the Haidian district (about 36m above the ground), which located in the northwest
urban area of Beijing, outside the third-ring road (39°56'N, 116°17'E). The sampling
site was located next to a high-density residential area, which has no significant
emissions from industrial in the surrounding neighborhood. Therefore, the observation
data could represent the air quality levels of the typical urban area of Beijing. The
observations were conducted in three seasons, 12nd Jan. to 14th Feb. for winter, 6th Jul.
to 21st Aug. for summer, and 30th Sep. to 13th Nov. for autumn.

**2.2 Instruments**

The $f$(RH) was measured by a dual-nephelometer system, one nephelometer for the
aerosol scattering coefficient under dry condition, and another nephelometer for the
humidified aerosol. The air flow first passed through a $PM_{2.5}$ inlet, and then was dried
by two tandem Nafion dryers (MD-700), which could reduce the RH of air flow lower
than 30%. The dried air was separated into two paths, one stream went directly into a
nephelometer, another stream was humidified by passing through a Gore-Tex tube,
which was set in a stainless steel tube. The interlayer between these two tubes is
circulating water headed by the water bath. The minutely scattering coefficients of dry
and humidified $PM_{2.5}$ under three wavelengths (450, 525, and 635nm) were
synchronously measured by these two nephelometers (Aurora 3000). In most studies,
there was only one water bath been used for humidifying. After a humidifying process,
it is necessary to wait for the water in the water bath to cool down enough for the next
process. Differently, in this study, two water baths were used. When one water bath was
heating up the water for humidifying, another water bath was cooling down the water
itself. After a humidifying process, the water bath with cool water would be switched
into the humidification pipeline. The use of two water baths could ensure that the
effective data of $f$(RH) is more than twice that of using only one water bath. The
temperature of water in the water bath was controlled by an automatic system to ensure
the humidifying effect.
Two combined RH and temperature sensors (Vaisala HMP110) were set at the inlet
and outlet of the wet nephelometer, respectively. The vapor pressures were calculated
by the sensor data, and the average value was considered as the vapor pressure in the
optical chamber. Thus, the humidified RH in the chamber could be calculated through
the derived vapor pressure and the temperature measured by the sensor in the chamber.




As mentioned above, the RH could not be accurately measured by a sensor when it is
above 90%. In order to accurately obtain higher relative humidity, the optical chamber
of wet nephelometer was deliberately cooled with the temperature lower than that at the
inlet and outlet. So the humidified RH could be higher than 95% in the chamber when
the RH at the inlet and outlet were lower than 90%. This method makes it possible to
observe the $f(RH)$ under high RH. To avoid the vapor condensation and particle
activation, the upper limit of humidified RH in the optical chamber of wet nephelometer
was set to 97%, which made the effective data of $f(RH)$ could reach RH of 96%. Each
humidifying process lasted about 50 minutes, and all the minutely average data were
automatically recorded by the control system. During the observation periods, these
two nephelometers were calibrated every ten days. Since two Vaisala sensors and two
nephelometers were all newly purchased and the relative error of vapor pressures was
always less than 1%, the sensors were not calibrated during the observation.
Other than the $f(RH)$ measurement, the six-minute average $PM_{2.5}$ mass
concentrations were also measured by a continuous dichotomous ambient air monitor
(TEOM 1405DF). The sample filter and sample conditioner filter of this monitor were
replaced every 15 days or when the dust loading exceeded 70%. The five-minute
average absorption coefficient of $PM_{2.5}$ was monitored by a multiangle absorption
photometer (MAAP 5012). The quartz-fiber filter could be automatically changed when
the light transmission was less than 20%. In addition, the hourly water-soluble ions
($SO_4^{2-}$, $NO_3^-$, $Cl^-$, $NH_4^+$, $Na^+$, $K^+$, $Mg^{2+}$, and $Ca^{2+}$) of $PM_{2.5}$ and trace gases (HCl, $HNO_3$,
$HNO_2$, $SO_2$, and $NH_3$) were measured by an online analyzer (MARGA). In addition,
the aerosol number concentration distribution (SMPS3938+APS3321), and size-
resolved chemical compositions (MOUDI 122) were also synchronously measured
during these three observation periods.
In this paper, we mainly focus on the discussions of $f(RH)$. In the near future, the
aerosol hygroscopicity would be comprehensively evaluated by making a use of all the
results from the above-mentioned observations and would be published in the following
papers.

**2.3 Methods**

The scattering Ångström exponent (Ångström, 1930) between 450 and 635nm ($Å_{450-}$
$_{635}$) at dry condition (RH<30%) characterizes the wavelength dependence of aerosol
scattering coefficients and was calculated using the scattering coefficient at
wavelengths of 635 and 450 nm by the following equation:

204         $Å_{450-635} = (\log \sigma_{sp}(450) - \log \sigma_{sp}(635)/(\log 635 - \log 450)$          (1)

The $PM_{2.5}$ absorption coefficient was measured in the wavelength of 670nm by MAAP.
In order to facilitate comparison, we transform the absorption coefficient of 670nm into
that of 525nm according to the assumption that absorption is inversely proportional to
wavelength (Bond and Bergstrom, 2006; Liu et al., 2018). Thus, the SSA at 525nm
($SSA_{525nm}$) was the proportion of the scattering coefficient to the sum of scattering
coefficient and absorption coefficient.
When the air is very clean, the relative change and fluctuation of the particle
concentration in the ambient air would be more intense. Furerthmore, the airflow into
two nephelometers could not be completely synchronized during the observation. The



larger relative error of scattering coefficient in two nephelometers could lead to greater
fluctuations in $f$(RH) during the humidifying cycle when under clean conditions
because the $f$(RH) is the ratio of the scattering coefficients from humidified and dry
nephelometers. Thus, the $f$(RH) points with dry scattering coefficient at 525nm less than
50 were removed from the fitting of the $f$(RH) curves. In this paper, the $f$(RH)
discussions are all based on the data of 525nm if not specifically pointed out.
The PM$_{2.5}$ concentrations are classified into three groups with 0~35 μg m$^{-3}$, 35~75
μg m$^{-3}$, and >75 μg m$^{-3}$, representing very clean, moderately clean, and polluted
conditions referring to the AQI grading standard of China, respectively. The reason for
this division is mainly based on the characteristics of $f$(RH) data.

## 3. Results and discussions

### 3.1 Overview of the optical properties and $f$(RH) of PM$_{2.5}$

Fig. 1 shows an overview of the hourly averaged light scattering coefficients
($\sigma_{sca,525nm}$), absorption coefficients ($\sigma_{ap,630nm}$), single scattering albedo ($SSA_{630nm}$), and
scattering Ångström exponent ($Å_{450-635}$) as well as $f$(RH) at RH=80% ($f$(80%)) for PM$_{2.5}$.
The average values of optical parameters and $f$(80%) in different seasons and under
different PM$_{2.5}$ pollution levels are listed in Table 1.
The PM$_{2.5}$ pollution was heaviest in the winter observation period and lightest in
summer. The scattering coefficient and absorption coefficient also show the same trends.
Single scattering albedo is one of the most important parameters in estimating of the
direct aerosol radiative forcing. The SSA$_{525}$ increased with the aggravation of PM$_{2.5}$
pollution in all three seasons, indicating that the components with strong scattering
ability, such as secondary ions, increased significantly during the pollution process. The
wind rose of SSA$_{525}$ in Figure 2 also indicates that the higher SSA$_{525}$ values generally
occurred under the southerly wind condition which was often accompanied by higher
PM$_{2.5}$ concentrations.
Scattering Ångström exponent is generally regarded as an indicator of particle size.
In winter, lower Å$_{450-635}$ was observed in less polluted conditions. As depicted in Figure
2, the clean conditions in winter occurred mainly in the case of northwest wind with
relatively higher wind speed, which led to a greater proportion of larger particles such
as crustal dust in the air. Conversely, the Å$_{450-635}$ in polluted condition was lowest in
summer. Compared with winter, the relative humidity in summer was much higher,
especially under the condition of pollution, which could make particle collision and
coagulation easier to occur Guo et al (2014).
According to the ZSR (Zdanovskii-Stokes-Robinson) assumption (Zdanovskii, 1948;
Stokes and Robinson, 1966), the $\kappa$ value of a multicomponent particle is equal to the
volume weighted average of each component. As depicted in Figure 1, the $f$(80%) at
525nm is highly correlated with the fractions of all the water-soluble ions to PM$_{2.5}$.
Owing to the proportion of hygroscopic components increased, PM$_{2.5}$ had higher $f$(80%)
in the polluted conditions. The standard deviations (SD) of $f$(80%) also indicate that the
changes of fractions of hygroscopic components were relatively small when the PM$_{2.5}$
over 35μg m$^{-3}$ in summer and autumn. The wind roses obviously reveal the differences
in the hygroscopicity and chemical compositions of PM$_{2.5}$ from different directions.
The $f$(80%) values in this study are in agreement with the range of values reported in





some other studies of the North China Plain (Table 2). Overall, the diurnal variation of
*f*(80) is not obvious, and the average *f*(80) at 12 to 16 pm was slightly higher (Figure
260   3).
262                                   **Table 1**
264                                   **Table 2**
266                                   **Figure 1**
268                                   **Figure 2**
270                                   **Figure 3**
**3.2 Parameterization schemes of *f*(RH)**
To better describe the dependence of *f*(RH) on RH, many different empirical
expressions have been applied in previous studies to fit the *f*(RH) measurements.
Kotchenruther et al. (1999) proposed that different fitting equations should be used
according to the observed curve structure. For monotonic curves in which *f*(RH) varies
smoothly with RH, they proposed the use of equation reported by Kasten (1969) and its
variants. The most commonly used equation is the one-parameter fit equation (Hänel,
1980; Gassó et al., 2000; Brock et al. 2016 ) and (Kotchenruther and Hobbs, 1998;
Carrico et al., 2003; Zieger et al., 2011; Chen et al., 2014). For deliquescent curves,
Kotchenruther et al. (1999) introduced a more complex equation, and more detailed
information and fitting equations could be found in Titos et al. (2016).
In this work, four commonly used empirical parameterization schemes were chosen
to describe the monotonic curves of *f*(RH) variation:
$f(RH)=a\ (1-RH/100)^{-\gamma(RH/100)}$    Chen et al. (2014)                    (2)
$f(RH)=a\ (1-RH/100)^{-\gamma}$    Kasten (1969)                    (3)
$f(RH)=1+a\ (RH/100)^{\gamma}$    Kotchenruther and Hobbs, (1998)                    (4)
$f(RH)=1+a\ (RH/(100-RH))$    Brock et al. (2016)                    (5)
where $\gamma$ parameterizes the magnitude of the scattering enhancement, which is not
affected by the RH. The comparison of the fitting results for different expressions is
shown in figure S1 to S3. According to the fitting results, $R^2$, and the comparison
between fitting *f*(80%) values and measured ones, we finally choose the Eq. (2) to
describe the scattering enhancement due to monotonic hygroscopic growth.
295                                   **Figure 4**
297                                   **Figure 5**
299                                   **Figure 6**
301                                   **Table 3**






In our previous studies, the pollution of particulate matter was usually classified into
three categories through two threshold values of $PM_{2.5}$ concentrations, $75\mu g \cdot m^{-3}$ and
$150\mu g \cdot m^{-3}$, for clean, moderately polluted, and heavily polluted conditions. In this study,
we found that the scatter points of $f$(RH) were quite concentrated near the fitting curves
when the $PM_{2.5}$ was above $75\mu g \cdot m^{-3}$, and the fitting parameters were markedly different
for conditions of $PM_{2.5}$ above and under $35\mu g \cdot m^{-3}$. Consequently, we use $35\mu g \cdot m^{-3}$ and
$75\mu g \cdot m^{-3}$ as boundaries to classify the different pollution levels in this study. Fig. (4) to
Fig. (6) show the fitting $f$(RH) curves under very clean, moderately clean, and polluted
conditions for three seasons. Table 3 shows the fitting results from this work and a
previous study conducted in NCP, which used the same equation expression (Chen et
al., 2014). Except for the very clean condition in winter, the fitting values of $a$ are all
near 1.0 and the fitting $f$(RH) curves are all above 1.0 for RH above 30%. However, the
fitting $f$(RH) curve shown in Fig 4a is apparently under the observed data points and
lower than 1.0 when RH under 40%. It means that the fitting curve of the very clean
condition in winter would underestimate the $f$(RH) value when RH is below 40%. It
can be seen from Figure 5 and Figure 6 that the fitting $R^2$ increases and the points
become more concentrated along with the aggravation of $PM_{2.5}$ pollution in summer
and autumn. This is in accordance with the characteristics of the SD of $f$(80) in Figure
3. Differently, the $f$(RH) points are still dispersed even under the polluted condition of
winter. This indicates that the chemical compositions of $PM_{2.5}$ when under the condition
of pollution were more stable in summer and autumn.
By comparing the fitting curves of different seasons and different pollution
conditions, it is found that the $f$(RH) curve of the very clean condition is lower than that
of other two pollution levels for each season. Except for the very clean condition in
winter and autumn, the fitting curves for the other seven conditions are relatively close,
especially when RH less than 90%. In the study of Chen et al. (2014), a segment fitting
with the critical RH of 60% is applied in the parameterization of $f$(RH). Comparing the
fitting results in Table 3, we find that the $f$(RH) values with RH above 80% from fitting
curves for clean and polluted conditions in Chen's study are evidently lower than those
from the respective curves for very clean and polluted conditions in this work. It
indicates that the scattering enhancement due to moisture uptake or hygroscopicity of
aerosol under high RH is probably higher than before.
In addition, the averaged $f$(RH) at 450nm, 525nm, and 635nm was also fitted
separately for each season. It is clear that the $f$(RH) showed a dependence on
wavelength, especially in summer. The averaged $f$(RH) increased with increasing
wavelength. Similar results were also obtained by Zhang et al. (2015) at Lin'an, China
and Zieger et al. (2014) at a regional continental research site in Melpitz, Germany.
However, we found that when under very clean conditions in winter and autumn, the
mean value of $f$(80) at 450nm was higher than that at 525nm and 635nm (Fig. S4), and
it is more obvious in winter. Through further curve fitting of $f$(RH) at different
wavelengths for three seasons, it is also found that the $f$(RH) curve of 450nm was
evidently higher than that of 525nm and 635nm only under very clean conditions in
winter. Our previous work showed that the sulfates, nitrates, and ammonium (SNA)



were abundant in aerodynamic diameter of 0.18~1.0μm on clean days in winter with
the mass median diameters (MMD) of SNA at about 0.45μm. However, the SNA was
mainly concentrated in 0.32~1.8μm with evidently higher MMDs on polluted days or
in other seasons (Zhao et al., 2017; Su et al., 2018). The high fraction of SNA in
particles below 500 nm might be responsible for the higher $f$(RH) at 450nm when under
very clean conditions in winter.
**3.3 Uncertainty analysis for $f$(RH) measurements**

353       As mentioned above, the humidified RH in the wet nephelometer was calculated
through the derived average vapor pressure and the temperature measured by the sensor
in the chamber. According to the differences in vapor pressure values from the sensors
at the inlet and outlet, a relative error of 0.5% could be calculated for the vapor pressure
data. And the mean absolute error of the temperature measurement in the nephelometer
was 0.2℃. Then, a Monte Carlo simulation was utilized to estimate the uncertainty of
calculated humidified RH values. New values of vapor pressure and temperature were
simulated by adding the uncertainties of 0.5% and 0.2℃ to the observation data
(following a normal distribution). And new humidified RH could be calculated using
the simulated data, and this procedure was repeated 1000 times for each humidified RH
value. Then the average standard error of humidified RH in the wet nephelometer could
be calculated to be 0.85%.

365       Next, the Monte Carlo simulation was used again. The RH was assumed to range
from 20% to 90% with steps of 1**,** and assuming that γ ranges from 0 to 1 with steps of
0.01. The chosen interval for γ covers the particle types from non-hygroscopic aerosol
particles to very hygroscopic particles. Therefore, more than 7000 conditions were
simulated, and each condition corresponded to one set of RH and γ. For each condition,
the dry scattering coefficients with a wide range of 1 to 1000 Mm$^{-1}$ were selected 5000
times as random numbers to present different atmospheric situations and aerosol loads.
The wet scattering coefficients were calculated associated with the previously selected
dry scattering coefficient and the $f$(RH) calculated by Eq. (2). A random error
(following a normal distribution with a standard variation of 0.85%) was also added to
the RH and the parameter $a$ was set 1.0 when calculating the $f$(RH). According to the
manual of nephelometer Aurora 3000, the standard error of aerosol scattering
coefficient is 2.5%. Then we simulated the dry and wet scattering coefficients by
assuming that they both had an uncertainty of 2.5% (following a normal distribution).
Thus, the simulated $f$(RH) can be calculated again.

380       The mean and relative standard deviation of simulated $f$(RH) were calculated for each
RH and γ and are shown in Fig. 7. For aerosols in this work (γ ~ 0.35), $f$(RH) errors
were below 4% with RH lower than 85% while reached 9.7% when RH= 96%, which
can be regarded as a conservative estimation. And other unpredictable factors
contributing to $f$(RH) uncertainty have not been considered in this approach.

**Figure 7**

**4. Conclusions**

388       A wide-range (30%-96%) and high-resolution humidified nephelometer system was
developed and a measurement campaign was conducted to study the $f$(RH) for three





seasons in 2017. The $f$(RH) at higher RH had firstly been monitored and reported.
Based on the overview of the optical properties and $f$(RH) of PM$_{2.5}$, we found that
the higher SSA$_{525}$ values generally occurred under the southerly wind components with
higher PM$_{2.5}$ concentrations. The $f$(80%) at 525nm of PM$_{2.5}$ was evidently higher under
the polluted conditions and highly correlated with the fractions of all the water-soluble
ions. The average $f$(80) at 12 to 16 pm was slightly higher than that of other periods
from the average diurnal variations.
By comparing the fitting results and curves of four different empirical
parameterization schemes, it was found that one of the two-parameter fit equations can
better fit the observed $f$(RH) data. The fitting curves could be widely applied to the
studies of atmospheric visibility, radiative force, or liquid water content due to aerosol
moisture absorption. For summer and autumn, the $f$(RH) points of polluted conditions
were more concentrated near the fitting curves in the scatter plots with higher fitting $R^2$.
And the fitting curve under the very clean condition was lower than that of other
conditions for each season. Compared with the fitting curves in the previous study, the
hygroscopicity of aerosol under higher RH has probably been enhanced. In summer,
the fitting $f$(RH) showed a significant dependence on wavelength and increased with
increasing wavelength for each pollution condition. Nevertheless, the $f$(RH) curve of
450nm was evidently higher than that of 525nm and 635nm under the very clean
condition in winter, due to the higher fraction of SNA in particles below 500 nm.
The uncertainties of $f$(RH) were simulated by considering all the predictable
uncertainties or errors during the measurement, which was below 10% for RH up to
412   96%.


*Data availability.* All data in this work are available by contacting the corresponding
author P. S. Zhao (pszhao@ium.cn).

*Author contributions.* P Z designed and performed this study. P Z and J D analyzed the
data and discussed the results. P Z prepared the manuscript and J D prepared all the
figures. X D and J S calibrated the device and collected the data.

*Competing interests.* The authors declare that they have no conflict of interest.

*Acknowledgments.* This work was supported by the National Natural Science
Foundation of China (41675131), the Beijing Talents Fund (2014000021223ZK49), the
Beijing Natural Science Foundation (8131003). We would also like to thank Dr. Ye
Kuang who was at Peking University for the help in the building of the nephelometer
system. Special thanks to the Max Planck Institute for Chemistry and Leibniz Institute
for Tropospheric Research where Dr. Zhao visited as a guest scientist in 2018.

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



## Table captions:

Table .1 Average $\sigma_{sca,525nm}$(dry), $\sigma_{abs,525nm}$(dry), $SSA_{525}$(dry), $\text{Å}_{450-635}$(dry), and $f$(80%)
under different pollution levels in three seasons.
Table 2. Comparisons of average $f$(80%) in different campaigns of NCP area.
Table 3. Comparisons of fitting parameters with another study using the same scheme.








Table 1

| | | Entire observation periods | PM$_{2.5}$ pollution levels ($\mu$g m$^{-3}$) | | |
|---|---|---|---|---|---|
| | | | Very clean (PM$_{2.5}\leq$35) | Moderately clean (35 <PM$_{2.5}\leq$75) | Polluted (PM$_{2.5}$>75) |
| Winter | $\sigma_{sca,525nm}$(dry) | 287±359 | 35±19 | 205±69 | 504±158 |
| | $\sigma_{abs,525nm}$(dry) | 46±37 | 10±7 | 41±13 | 75±15 |
| | SSA$_{525}$(dry) | 0.82±0.06 | 0.79±0.04 | 0.83±0.03 | 0.87±0.02 |
| | Å$_{450-635}$(dry) | 1.27±0.20 | 1.18±0.13 | 1.42±0.15 | 1.45±0.15 |
| | f(80%) | 1.47±0.16 | 1.31±0.12 | 1.51±0.09 | 1.60±0.14 |
| Summer | $\sigma_{sca,525nm}$(dry) | 170±125 | 85±46 | 226±87 | 410±121 |
| | $\sigma_{abs,525nm}$(dry) | 27±13 | 20±10 | 30±11 | 40±12 |
| | SSA$_{525}$(dry) | 0.83±0.09 | 0.79±0.09 | 0.87±0.05 | 0.91±0.02 |
| | Å$_{450-635}$(dry) | 1.34±0.23 | 1.42±0.22 | 1.30±0.22 | 1.18±0.21 |
| | f(80%) | 1.54±0.16 | 1.50±0.19 | 1.60±0.08 | 1.62±0.06 |
| Autumn | $\sigma_{sca,525nm}$(dry) | 261±243 | 57±54 | 241±62 | 564±169 |
| | $\sigma_{abs,525nm}$(dry) | 38±24 | 17±12 | 40±16 | 61±18 |
| | SSA$_{525}$(dry) | 0.83±0.09 | 0.75±0.09 | 0.85±0.05 | 0.90±0.03 |
| | Å$_{450-635}$(dry) | 1.17±0.27 | 1.16±0.27 | 1.30±0.20 | 1.06±0.27 |
| | f(80%) | 1.53±0.11 | 1.44±0.14 | 1.58±0.08 | 1.57±0.04 |


Table 2

| Study area (Campaign) | Periods | Aerosol pollution levels | f(RH=80%) | Wavelength (nm) | Reference |
|---|---|---|---|---|---|
| The rural site of Beijing | 24 April–15 May 2006 | Clean | 1.31 ± 0.03 | 525 | Pan et al. (2009) |
| | | Urban pollution | 1.57 ± 0.02 | | |
| | | Special case | 2.21 | | |
| SDZ, Beijing, a rural site | December 2005 | Relatively clean | 1.16 | 525 | Yan et al. (2009) |
| | | Relatively polluted | 1.34 | | |
| CAMS, Beijing, an urban site | | Relatively clean | 1.2 | 525 | |
| | | Relatively polluted | 1.48 | | |
| Wuqing, Tianjin | October 2009 to late January 2010 | Clean | 1.46 ± 0.15 | 550 | Chen et al. (2014) |
| | | Polluted | 1.58 ± 0.19 | | |
| Wangdu, suburban district of North China Plain | 4 June 2014 - 14 July 2014 | Entire campaign | 1.8 (1.1-2.3) | 550 | Kuang at al. (2017) |
| | | deliquescent phenomena | 2.0 (1.7-2.3) | | |






Table 3

| | Periods | | $a$ | $\gamma$ | Reference |
|---|---|---|---|---|---|
| Very clean | | | 0.930 | 0.329 | |
| Moderately clean | 12 Jan.-14 Feb., 2017 | | 0.971 | 0.372 | |
| Polluted | | | 0.988 | 0.356 | |
| Very clean | | | 0.972 | 0.355 | |
| Moderately clean | 6 July-21 Aug., 2017 | | 0.980 | 0.362 | This work |
| Polluted | | | 0.984 | 0.371 | |
| Very clean | | | 0.979 | 0.334 | |
| Moderately clean | 30 Sep. to 13 Nov., 2017 | | 1.002 | 0.344 | |
| Polluted | | | 1.014 | 0.332 | |
| Entire campaign | | RH<60% | 1.02 | 0.21 | |
| | | RH≥60% | 1.08 | 0.26 | |
| Clean | Oct.2009- | RH<60% | 1.00 | 0.10 | Chen et al. |
| | Jan.2010 | RH≥60% | 1.00 | 0.26 | (2014) |
| Polluted | | RH<60% | 1.03 | 0.26 | |
| | | RH≥60% | 1.14 | 0.25 | |





## Figure captions:

Figure 1. Time series of $\sigma_{sca,525nm}(dry)$, $\sigma_{abs,525nm}(dry)$, $SSA_{525}(dry)$, $Å_{450-635}(dry)$, $f(80\%)$, and mass concentrations of water-soluble ions and their mass fractions over all the sampling periods in three seasons.

Figure 2. Wind dependence of $\sigma_{sca,525nm}(dry)$, $\sigma_{abs,525nm}(dry)$, $SSA_{525}(dry)$, $Å_{450-635}(dry)$, and $f(80\%)$ over three seasons; the shaded contour indicates the average of variables for varying wind speeds and wind directions.

Figure 3. Diurnal variations of $f(80\%)$ in different seasons.

Figure 4. Fitting $f(RH)$ curves under different pollution levels in winter.

Figure 5. Fitting $f(RH)$ curves under different pollution levels in summer.

Figure 6. Fitting $f(RH)$ curves under different pollution levels in autumn.

Figure 7. Simulated $f(RH)$ and its error (color scale) as a function of RH and the hygroscopic parameter $\gamma$.








Figure 1




Figure 2








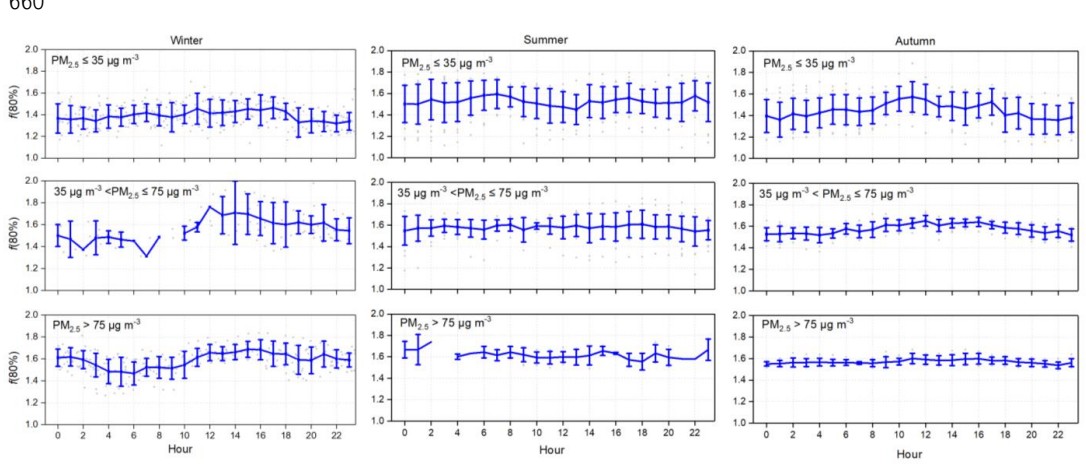


Figure 3

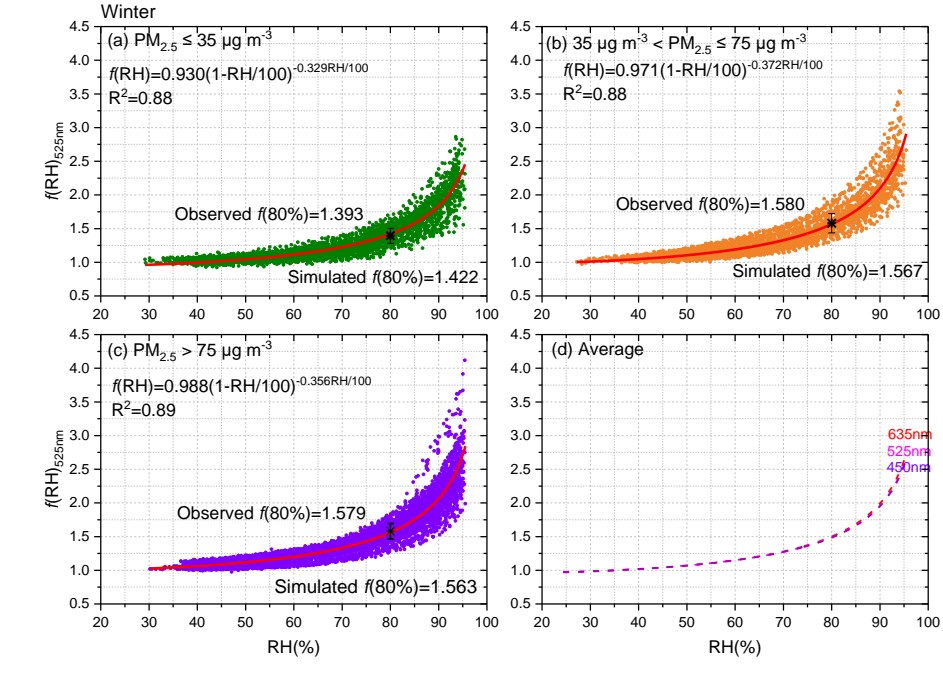



Figure 4



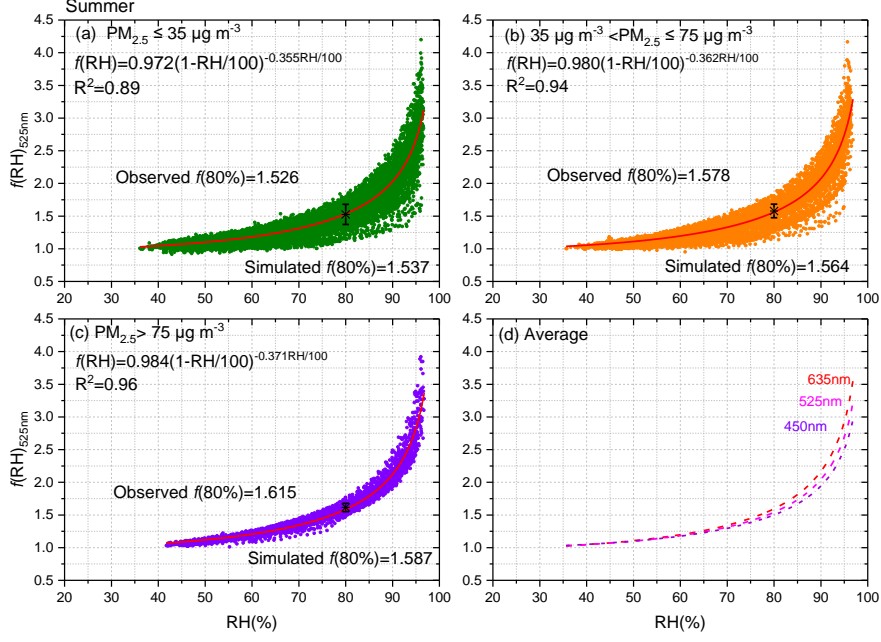



Figure 5


Figure 6





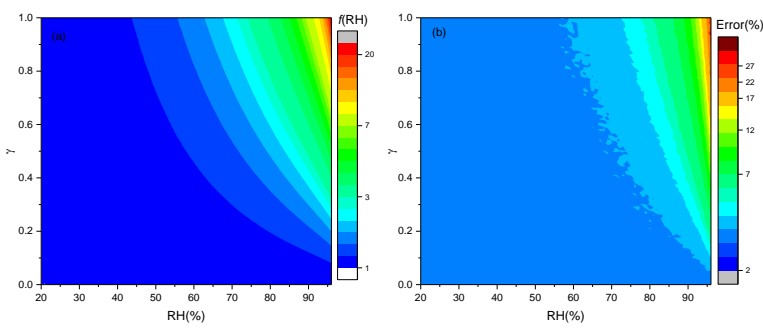


670                                                  Figure 7
