# Peer review of "Hygroscopic growth effect on aerosol light scattering in the urban area of"

_Atmospheric Chemistry and Physics, 2018_

## Referee Comment (RC1) · Anonymous Referee #1 · 14 Sep 2018

Review of Zhao et al. (2019) submitted to Atmos Chem Phys.

Summary The paper examines the hygroscopic response of aerosols in the North China Plain in the urban area of Beijing.

The results should generate sufficient interest and provide useful information for the aerosol community. The paper is suitably referenced. The paper needs further work with English and grammar as well as significant rewriting of the analysis to make it more impactful.

[Figure]

Technical Comments The authors have about 1 month of data for 3 seasons in one year. I would hardly call this a long term measurement campaign as in the title.

I find no obvious technical flaws in assessing the literature and discussing their f(RH) measurements. However, calibration protocols are not discussed at any length. Some more description of calibration of the nephelometers is merited. Was it $CO_2$ and clean air? What were the changes over time? The DeBell paper referenced is not a suitable reference for climate effects of aerosols. It's a report on visibility effects.

There were some early measurements of aerosol hygroscopicity in China by Xu, Bergin et al. in the early 2000's that are worth integrating into this paper.

35 $\mu$g/m3 is hard to fathom as 'very clean' conditions. This is the 24 hr PM2.5 standard limit in the US (while the annual standard is 12). I suggest renaming or adding a disclaimer that these definitions are not globally applicable and vary by country. It would be useful to indicate the fraction in time in each of the three PM2.5 levels.

Line 109, I'm not sure that everyone knows the NCP is the most severe area of pollution in China. Does it exceed the Pearl River Delta and other heavily industrialized regions?

'moisture absorption ability' replace with water uptake as aerosols can both absorb and adsorb water. Can you estimate the residence time of the humidified flow with respect to achieving equilibrium? Line 218 Less than 50 1/Mm?

Line 240 discussion of angstrom is hard to follow as to what conditions (season, pollution level etc) dominated when. Suggest a rewrite to make more clear. The discussion of results in lines 300-350 also is somewhat long and hard to comprehend.

Low SSA during clean conditions, to what is this attributed? A large elemental carbon fraction is suggestive of soot from urban-industrial or possibly biomass combustion.

The Monte Carlo simulation of RH uncertainty is interesting. So how does this relate to the typical +/-2

Mechanics and Presentation The writing needs a fair bit of work in terms of clarity, grammar and English. The instruments section 2.2 in particular. I cannot give all the polishing needed but will illustrate with a suggested rewriting of the abstract. The abstract really does not summarize the key points of the paper adequately and thus I have rewritten to also improve the English.

"Hygroscopicity is a critical property of ambient aerosols that affects atmospheric light extinction, aerosol radiative forcing of climate, and the formation processes and sinks of aerosols. The light scattering hygroscopic growth factor (f(RH)) quantifies this for polydisperse ambient aerosols for continuous relative humidity (RH), typically over the range 20

Line 67, suggest starting a new paragraph with f(RH).

Line 155, suggest the form here and elsewhere: 'nephelometers (Ecotech, Inc., Model Aurora 3000)' Figure 1, the seasonal panels share the same scale which is helpful, with the exception of the light scattering coefficient time series.

Does it make sense to have a supplement for a single figure? This could be incorporated into the main paper without much 'cost'.

Table 1 units should be specified for scattering and absorption, presumably 1/Mm The conclusion ends rather abruptly.

---

## Referee Comment (RC2) · Anonymous Referee #2 · 13 Jan 2019

Review of "Hygroscopic growth effect on aerosol light scattering 1 in the urban area of Beijing: a long-term measurement by a wide-range and high-resolution humidified nephelometer system" by Zhao et al.

This study reports on ambient f(RH) measurements in Beijing, China. The general topic of aerosol hygroscopicity is of interest to readers of this journal. This work definitely requires English editing if there is a subsequent version. The manuscript is sloppy with many writing errors that I did not fully outline below but should be fixed. The main issue of this work is the lack of novelty in the scientific findings. There is a general lack

of depth in the analysis and the paper reads like a very brief lab report currently that still lacks many easily reportable basic descriptive statistics of their data. The paper is disorganized with an example of this being that Section 3.3 is disjointed from the rest of the paper. The authors are encouraged to review the literature better and to examine their dataset more deeply to find novel results that would be of broad interest to readers rather than being a quick report of values very specific to their region. In my view, to make this paper reach the level of quality ACP warrants, the authors should use the other datasets they advertised they would use in Lines 196-198.

Specific Comments: Line 47: remove the word "the"

Line 64-66: Numerous techniques can measure g(RH) and not just the HTDMA. Authors should mention other techniques used to measure g(RH) in field studies.

Line 109: Remove "As we all know" since we all may not know as well as the authors about that region.

Line 395: 16 pm doesnt make much sense. use military time

Figure 1: too small to read anything in the panels.

---

## Author Comment (AC1) · 19 Feb 2019

**Summary The paper examines the hygroscopic response of aerosols in the North China Plain in the urban area of Beijing.**
**The results should generate sufficient interest and provide useful information for the aerosol community. The paper is suitably referenced. The paper needs further work with English and grammar as well as significant rewriting of the analysis to make it more impactful.**

Thank you very much for your comments. We seriously revised the parts of this paper that were not clear enough or not necessary, and we also rewrote the analysis parts following your advices. In addition, we asked a professional English editing website to revise our paper for grammar and expression. Furthermore, in order to enrich the content and highlight the key points, we added an analysis of f(RH) under high relative humidity conditions.

**Technical Comments The authors have about 1 month of data for 3 seasons in one year. I would hardly call this a long term measurement campaign as in the title.**
**I find no obvious technical flaws in assessing the literature and discussing their f(RH) measurements. However, calibration protocols are not discussed at any length. Some more description of calibration of the nephelometers is merited. Was it CO2 and clean air? What were the changes over time? The DeBell paper referenced is not a suitable reference for climate effects of aerosols. It's a report on visibility effects.**

This measurement campaign was conducted from July to Nov. for summer and autumn. In order to make the length of observation periods of three seasons similar, we selected representative data of more than one month for each season. As this reviewer said, it is hardly to call this a long term measurement campaign. So we remove the word "long-term" in the title. In the manual for Aurora 3000, it is required that a full calibration only needs to be performed approximately every 3 months. Since the relative humidity in the optical chamber was always changing, during the measurement, a full calibration was performed every ten days, and the R-134 was used as the span gas. In addition, the DeBell paper is really a report on visibility, and we change the position of this citation in the text.

**There were some early measurements of aerosol hygroscopicity in China by Xu, Bergin et al. in the early 2000's that are worth integrating into this paper.**

One paper of Xu and Bergin involving f(RH) measurement was introduced in the introduction section of the revised manuscript.

**35 g/m3 is hard to fathom as 'very clean' conditions. This is the 24 hr PM2.5 standard limit in the US (while the annual standard is 12). I suggest renaming or adding a disclaimer that these definitions are not globally applicable and vary by country. It would be useful to indicate the fraction in time in each of the three PM2.5 levels.**

According to the Technical Regulation on Ambient Air Quality Index of China, the air quality is classified to **Excellent** or **Superior** level when the 24-hour average of PM2.5 is below 35μgm$^{-3}$. There is no PM2.5 hourly concentration standard in China, so we refer to the 24-hour standard to determine the pollution level for hourly data. Compared with the 24-hour concentration standard, the criterion of hourly concentration below 35 is more stringent. We add this explanation to the revised manuscript.

**Line 109, I'm not sure that everyone knows the NCP is the most severe area of pollution in China. Does it exceed the Pearl River Delta and other heavily industrialized regions?**

The NCP is the most air polluted area in China, which should be a consensus in China. In recent years, more than half of the 10 cities with the worst air quality in China come from this region.

**'moisture absorption ability' replace with water uptake as aerosols can both absorb and adsorb water. Can you estimate the residence time of the humidified flow with respect to achieving equilibrium? Line 218 Less than 50 1/Mm?**

The 'moisture absorption ability' was replaced with 'water uptake'.

The airflow into the nephelometer was controlled by an orifice, which was about 5.0 l/min. The residence time of particles staying in the humidified tube and the optical chamber was more than 10s, which was enough for particles to reach equilibrium. When the air is clean, it is often accompanied by higher wind speed, which makes the relative changes of particle concentration and particle size distribution in the air more intense. In addition, the switching time between three measurement modes in two nephelometers could not be completely synchronized. This resulted in a relatively larger fluctuation of f(RH) curve for clean conditions. To avoid these unevaluable uncertainties, the f(RH) points with dry scattering coefficient at 525nm less than 50 Mm$^{-1}$ were removed from the fitting of the f(RH) curves.

**Line 240 discussion of angstrom is hard to follow as to what conditions (season, pollution level etc) dominated when. Suggest a rewrite to make more clear. The discussion of results in lines 300-350 also is somewhat long and hard to comprehend.**

According to your suggestion, we reorganized these two parts and revised the text that was not clear enough or not necessary.

**Low SSA during clean conditions, to what is this attributed? A large elemental carbon fraction is suggestive of soot from urban-industrial or possibly biomass combustion.**

The northern part of Beijing is close to the mountains and grasslands areas where is seldom affected by human activities. Therefore, most of the clean conditions in Beijing are accompanied by northerly wind. On clean days, the scattering and absorption coefficients of aerosols will be greatly reduced, but the concentration of secondary ions,

which has a greater impact on scattering, will decrease more significantly.

**The Monte Carlo simulation of RH uncertainty is interesting. So how does this relate to the typical +/-2**

I am not very clear about the specific meaning of this comment. In the revised version, this part is placed at the back of the text as an appendix.

**Mechanics and Presentation The writing needs a fair bit of work in terms of clarity, grammar and English. The instruments section 2.2 in particular. I cannot give all the polishing needed but will illustrate with a suggested rewriting of the abstract. The abstract really does not summarize the key points of the paper adequately and thus I have rewritten to also improve the English.**
**"Hygroscopicity is a critical property of ambient aerosols that affect atmospheric light extinction, aerosol radiative forcing of climate, and the formation processes and sinks of aerosols. The light scattering hygroscopic growth factor (f(RH)) quantifies this for polydisperse ambient aerosols for continuous relative humidity (RH), typically over the range 20**

Thank you very much for your kind suggestions. We carefully revised the words and rewrite the abstract. Further, we also asked a professional English editing website to polish the language.

**Line 67, suggest starting a new paragraph with f(RH).**

The contents of the introduction for f(RH) were put together in a new paragraph.

**Line 155, suggest the form here and elsewhere: 'nephelometers (Ecotech, Inc., Model Aurora 3000)' Figure 1, the seasonal panels share the same scale which is helpful, with the exception of the light scattering coefficient time series.**

The company name or brand of each instrument is added. The scales of ordinates in sub-figures for the light scattering coefficient in Figure 1 are changed to be uniform.

**Does it make sense to have a supplement for a single figure? This could be incorporated into the main paper without much 'cost'.**

In the supplementary document, there are four figures in all. The first three figures mainly show the results of f(RH) curve fitting with other different schemes. The last figure exhibits the differences between frequency distributions of f(RH80) at three wavelengths. These pictures are for further reference and are not suitable for placing in the main paper.

**Table 1 units should be specified for scattering and absorption, presumably 1/Mm**
The units of $Mm^{-1}$ for scattering and absorption are added in Table 1.

**The conclusion ends rather abruptly.**
The section of Conclusions was rewritten according to the changes in the text.

---

## Author Comment (AC2) · 19 Feb 2019

**This study reports on ambient f(RH) measurements in Beijing, China. The general topic of aerosol hygroscopicity is of interest to readers of this journal. This work definitely requires English editing if there is a subsequent version. The manuscript is sloppy with many writing errors that I did not fully outline below but should be fixed. The main issue of this work is the lack of novelty in the scientific findings. There is a general lack of depth in the analysis and the paper reads like a very brief lab report currently that still lacks many easily reportable basic descriptive statistics of their data. The paper is disorganized with an example of this being that Section 3.3 is disjointed from the rest of the paper. The authors are encouraged to review the literature better and to examine their dataset more deeply to find novel results that would be of broad interest to readers rather than being a quick report of values very specific to their region. In my view, to make this paper reach the level of quality ACP warrants, the authors should use the other datasets they advertised they would use in Lines 196-198.**

Thank you very much for your valuable comments. These are very helpful for the revision of this paper. Based on your suggestions, for this paper, we mainly made the following modifications and adjustments.

(1) The observation of f(RH) by humidified nephelometer has been carried out worldwide for many years, making it possible to obtain the hygroscopicity of aerosols under continuous relative humidity. Previously, f(RH) data were mainly used to evaluate the hygroscopicity and radiative forcing of aerosols. Recently, f(RH) was also used to calculate the hygroscopicity parameter, the aerosol water content, and the number concentrations of cloud condensation nuclei (Kuang et al., 2017, 2018; Tao et al., 2018) based on multi-wavelength observations. However, the enhancement of light scattering or growth of size for aerosols under high humidity (>90%) has been seldom reported due to the difficulty in creating a stable high humidity and measuring it accurately. In previous observations, the temporal resolution of f(RH) observation was usually low, and the results of f(RH) could not be obtained stably and accurately for high humidity. In order to meet the urgent needs of evaluation and forecast for low visibility and aerosol radiative forcing under heavy pollution in the North China Plain, an improved high-resolution humidified nephelometer system was established to observe the $f$(RH) of $PM_{2.5}$ for a wide RH range between 30%-96%. It was the first high-RH f(RH) observation by the humidified nephelometer system in China. The f(RH) data itself and its fitting parameterization can be directly applied to relevant research and specific work in the NCP.

Thus, we carefully revised the words and rewrote many parts of this paper to highlight the key points and make the analysis more impactful.

(2) To enrich the content and enhance the depth of analysis, we added a new section into the text for analyzing the impact of hygroscopic growth under high humidity on light scattering of PM2.5 and emphasize the importance of high humidity data for the overall hygroscopicity of aerosols and the f(RH) curve fitting. We found that the

enhancement of light scattering for PM2.5 would be overestimated without the high humidity data. In addition, based on the MARGA data, we also evaluated the quantitative relationship between water-soluble components and hygroscopicity.

(3) In the revised version, the part of uncertainty analysis was placed at the back of the text as an appendix.

(4) We also rewrote the sections of abstract and conclusions to make them more focused and logical. Further, we asked a professional English editing website to polish the language.

**Specific Comments:**
**Line 47: remove the word "the"**
The word "the" was removed.

**Line 64-66: Numerous techniques can measure g(RH) and not just the HTDMA. Authors should mention other techniques used to measure g(RH) in field studies.**
The Humidifying-Differential Mobility Particle Sizer (Eichler et al., 2008; Meier et al., 2009) and the Differential Aerosol Sizing and Hygroscopicity Spectrometer Probe (DASH-SP) (Sorooshian et al., 2008), two instruments which are both based on the differential mobility of particles are added in Introduction.

**Line 109: Remove "As we all know" since we all may not know as well as the authors about that region.**
"As we all know" was removed.

**Line 395: 16 pm doesn't make much sense. use military time**
16 pm was changed to 16 o'clock.

**Figure 1: too small to read anything in the panels.**
Figure 1 was redrawn and enlarged.